

# Tagging Jets in Invisible Higgs Searches

Anke Biekötter*, Fabian Keilbach, Rhea Moutafis, Tilman Plehn,
and Jennifer M. Thompson

Institut für Theoretische Physik, Universität Heidelberg, Germany

* biekoetter@thphys.uni-heidelberg.de

## Abstract

Searches for invisible Higgs decays in weak boson fusion are a well-known laboratory for jets and QCD studies. We present a series of results on tagging jets and central jet activity. First, precision analyses of the central jet activity require full control of single top production in some analyses. Second, the rate dependence on the size of the tagging jets is not limited to weak boson fusion. For the first time, we show how subjet information on the tagging jets and on the additional jet activity can be used to extract the Higgs signal. The additional observables relieve some of the pressure on other, critical observables. Finally, we compare the performance of weak boson fusion and associated Higgs production.


## Content

# 1 Introduction

The discovery of a fundamental Higgs boson [1,2] guarantees the LHC a place among the great experimental efforts in science and establishes perturbative gauge theory as the framework that describes fundamental interactions in nature. Since then we have been experiencing an essential change in research performed at colliders. Quietly and against all odds, LHC physics has become a field of experimental and theoretical precision physics. This is closely related to a new way we think about QCD and jets. From being regarded as a nuisance in going from lepton colliders to hadron colliders, jets have become a powerful analysis object far beyond simply describing partons from hard processes. Key directions in jet physics include QCD radiations as well as jet sub-structure including advanced machine learning methods. This defines a turning point, where it is not obvious if by the end of the current run ATLAS and CMS should still use jets as contained objects rather than an optional link between jet-based observables and sub-structure observables.

In view of both its physics potential and these technical developments, one of the most interesting LHC analyses is the search for Higgs decays into invisible particles [3]. On the physics side, dark matter is the big open question in particle physics cosmology. In the Standard Model, Higgs decays to neutrinos are extremely rare. In models for physics beyond the Standard Model, invisible Higgs decays are a generic signature. One common structural element is the super-renormalizable Higgs mass term, which allows any singlet field to mix with the Higgs and which opens a renormalizable portal to a hidden sector [4]. This Higgs portal opens a wealth of options for model building ranging from simple dark matter models to more complicated unified models [5–7]. Whatever guides the exact composition of such a hidden sector, a Higgs portal would always show itself through an invisible decay width.

On the phenomenological and analysis side, there are various strategies to detect invisible Higgs decays at the LHC. The classic search strategy for invisible Higgs decays is based on weak-boson-fusion (WBF) Higgs production [8–10]. Two forward tagging jets [11, 12] can help with triggering, and the massive $W$-propagators automatically provide a minimum amount of missing energy from the Higgs decay. A general challenge in weak boson fusion processes are the large QCD backgrounds, which can be controlled through a comprehensive analysis of the tagging jet kinematics and the suppressed hadronic activity in the central detector. It is worth noting that the difference in jet structure of the WBF signal and the QCD backgrounds follows from first-principles perturbative QCD.

Formally of the same perturbative order as the WBF production process, but with a distinctive resonance structure, boosted Higgs production in association with either a $W$ or a $Z$ boson will significantly add to the LHC reach [13, 14]. A leptonic $Z$-decay not only guarantees triggering, but also provides a powerful handle in reducing all QCD backgrounds. On the other hand, the production rate in this channel is has a significantly smaller rate than WBF Higgs production. Finally, searches for invisible Higgs decays in $t\bar{t}H$ production [15] will be a challenge even at the high-luminosity LHC, both statistically and systematically.

Experimental searches for invisible Higgs decays by CMS rely on weak boson fusion [16], $ZH$ production [17], and a combination of both production processes [18,19]. Similarly, in ATLAS there are searches in weak boson fusion [20], $ZH$ production [21], and a combination including the hadronic $ZH$ channel [22]. The reach in terms of an invisible branching ratio of a Standard-Model-like Higgs boson ranges around 23%, but is expected to reach the 2 ... 3% level at the end of the high-luminosity run [9]. To avoid unnecessary assumptions about the Higgs production rate, an invisible Higgs search is best included in a global Higgs and gauge sector analysis [23]. An additional improvement to the sub-percent level can be reached in the

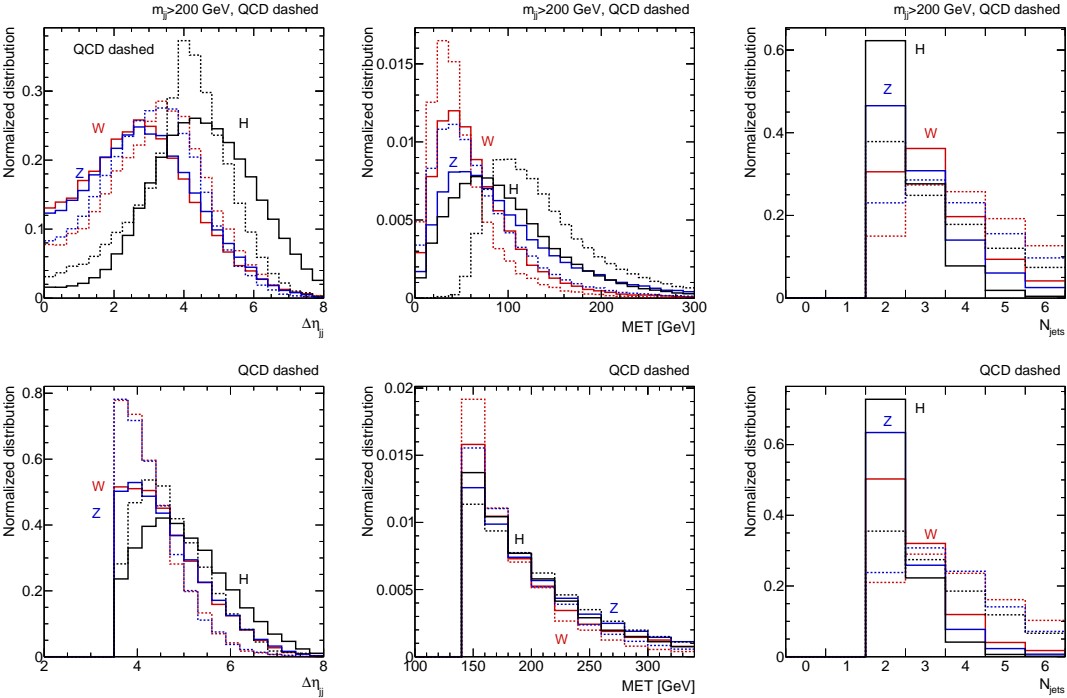

Figure 1: WBF signal and background distributions after the minimal requirement $m_{jj} > 200$ GeV (upper panels) and after the pre-selection cuts of Eq.(3). The $\Delta\eta_{jj}$ (left) distribution for the $V$+jets $Z$ backgrounds look very similar. Deviations between $W$ and $Z$ backgrounds appear for the missing energy distribution (center) and $N_{\text{jets}}$ (right).

same signatures at a future 100 TeV hadron collider [24]. Even more optimistic predictions for the 100 TeV hadron collider are typically based on ignoring the known leading sources of uncertainties.

In this study we focus on the WBF production channel and possible improvements through an improved understanding of the tagging jets and the central jet activity. In Sec. 2 we start by discussing some background issues which occur when we link the irreducible $Z$+jets backgrounds and the $W$+jets backgrounds, in particular the impact of single top production as a future key background. Next, we focus on the structure of the tagging jets, including the effects of a change in jet size (Sec. 3) and its quark vs gluon content (Sec. 4). The final, crucial question for this signature in the high-luminosity LHC environment is how much it gets degraded by stronger trigger requirements. In Sec. 5 we compare different triggering scenarios with a $ZH$ benchmark analysis.

## 2   V+jets backgrounds

The leading channel to search for invisibly decaying Higgs bosons is Higgs production in weak boson fusion [8, 9]. The reason is that the recoil against the tagging jets automatically leads to a transverse momentum $p_{T,H} \gtrsim m_W/2$ and that the tagging jet structure with a reduced central jet activity allows us to control QCD backgrounds [11]. In principle, this channel could be sensitive to invisible branching ratios down to $\text{Br}_{\text{inv}} \approx 2\% \ldots 3\%$ for integrated luminosities of 3 ab$^{-1}$. To test branching ratios below the percent range we will have to go to a 100 TeV

hadron collider [24, 25].

While the structure of the hard weak boson fusion process is relatively simple, the LHC reach in this channel is largely determined by our understanding of the backgrounds, including central-detector QCD features in the signal and background events. The main irreducible background is $Z$+jets production with a decay $Z \to \nu\bar{\nu}$. The nominally reducible $W$+jets production with an unobserved decay lepton can contribute at a similar level. The two hard jets of the signal can be produced either through a hard QCD process, $\sigma(Vjj) \propto \alpha_s^2\alpha$, or through a hard electroweak process $\sigma(Vjj) \propto \alpha^3$, with $V = Z, W$. It is crucial to separate these two production mechanisms if we later use the hadronic activity to separate the Higgs signal from those backgrounds.

The $Z$-background can be measured using the visible decays $Z \to \ell\ell$, but in a control sample which due to the $Z$ branching ratios is smaller than the actual background sample. The $W$+jets background with a leptonic decay $W \to \ell\bar{\nu}$ occurs when we lose the lepton in the detector. Aside from the corresponding phase space effects, preferring a forward or soft lepton, it should look very similar to the $Z$+jets background.

We simulate the WBF signal and its background processes for a 14 TeV high-luminosity LHC at LO using SHERPA2.2.1 [26] with up to three or four hard jets combined in the CKKW scheme [28]. For the matrix element, we employ COMIX [27]. For both backgrounds we separate the QCD and weak sub-processes. For the signal we also take into account the contribution from gluon fusion, denoted as QCD signal contribution [29]. The corresponding event sample is generated at LO with two hard jets using SHERPA, employing OPENLOOPS [30–33] to include the finite top mass effects in the loop. The two tagging jet candidates are defined as the two hardest anti-$k_T$ jets in the event using FASTJET [34] with a jet size $R = 0.4$. Detector effects are taken into account using DELPHES3.3 [35] and the ATLAS card with an updated lepton efficiency [36, 37].

In the upper panels of Fig. 1 we show the two $W$ and $Z$ backgrounds in comparison to the Higgs signal after the minimal requirement

$$m_{jj} > 200 \text{ GeV} . \tag{1}$$

Because the signal process includes the $ZH \to (jj)H$ topology, we need to apply this simple kinematic cut to select the WBF diagrams. For both the QCD and the weak sub-processes we observe that the $\Delta\eta_{jj}$ (and $m_{jj}$) background distributions look very similar for the $Z$+jets and $W$+jets backgrounds. The missing energy distribution is significantly softer for the $W$+jets background, both in the QCD case and in the electroweak case. Especially for the QCD process we do not expect any difference between the $Z$+jets and $W$+jets distributions before detector effects. The QCD $W$+jets process only contributes as a background if the lepton leaves sufficient tracks in the detector to be reconstructed. However, even though a lepton is not reconstructed as such it will still deposit energy in the calorimeter, leading to a reduced missing energy recoiling against the visible constituents in the detector. We have checked that our observed difference in the $\slashed{E}_T$ distribution is due to this effect and easily accounted for in experiment.

The moment we apply something like a central jet veto [8, 9], the number of jets in the signal and background events is a crucial observable. We show the number of jets with $p_{T,j} > 20$ GeV and $|\eta_j| < 4.5$ in the right panel of Fig.1. Instead of the expected similar behavior for the $Z$+jets and $W$+jets backgrounds, we find significantly more jets in electroweak $W$+jets events than in electroweak $Z$+jets events. Unlike the signal and all other backgrounds, the electroweak $W$ background is more likely to include three or more jets in an event. The

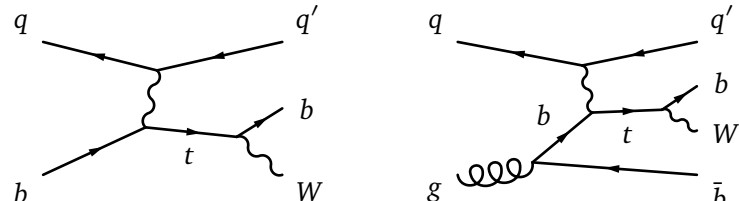

Figure 2: Example Feynman diagrams for the single top contribution to the $W$+jets background.

dominant effect that leads to this behavior is the contribution of single top process to this background category,

$$pp \to bW^+ + \text{jets}. \tag{2}$$

We display the corresponding Feynman diagrams in Fig. 2. For events with exactly two jets in the final state, the contribution from those diagrams is 30% for $m_{jj} > 200$ GeV. For three-jet events the single top process can contribute up to 50% of all events, eventually forcing us to understand this difficult process with high precision during the future LHC runs.

Before we enter any dedicated signal vs background analysis we need to apply a set of basic cuts, motivated by detector performance, triggering, and rejecting generic backgrounds. Following the CMS pre-selection [19] we require two tagging jets combined with sizeable missing transverse energy pre-selection

$$
\begin{array}{lll}
p_{T,j_{1,2}} > 40\,\text{GeV} & |\eta_{j_{1,2}}| < 4.5 & \not{E}_T > 140\,\text{GeV} \\
\eta_{j_1}\,\eta_{j_2} < 0 & |\Delta\eta_{jj}| > 3.5 & m_{jj} > 600\,\text{GeV} \\
p_{T,j_3} > 20\,\text{GeV} & |\eta_{j_3}| < 4.5 & \text{(if 3rd jet available)}\,.
\end{array} \tag{3}
$$

and veto any additional observed lepton with a transverse momentum larger than $p_{T,\ell} > 7$ GeV based on our fast DELPHES3.3 detector simulation. In the lower panels of Fig. 1 we show the corresponding distributions. The good news is that once we apply these pre-selection cuts the single top contamination in the $W$+jets sample drops to below 5% for two-jet events and 12% for three-jet events. Therefore, the QCD $V$+jets background resemble each other much more closely.

## 3 Tagging jet size

Traditionally, the jet size in a given ATLAS or CMS analysis is chosen following experimental considerations. However, as has been recently pointed out, the geometric size of the tagging jets in WBF Higgs production can have a sizeable impact on the effect of higher-order corrections to the rate [38]. Obviously, this is an effect of real parton emission and how partons in addition to the two hard tagging jets are combined into jets. For our simulations we use the same tool chain as in the previous Sec.2, combining up to three hard jets and parton shower radiation. We again define the tagging jets as the hardest two jets with $p_{T,j} > 20$ GeV using the anti-$k_T$ algorithm in FASTJET, but now with a variable jet size $R = 0.4 \ldots 1.0$.

We first show the $R$-dependence for the WBF signal

$$pp \to jj\,H_{\text{inv}} \tag{4}$$

in Fig. 3. This signal definition includes the $ZH$ topology, so we extract the WBF channel in a two-jet plus parton shower setup with no $p_{T,j}$ cut, but requiring $m_{jj} > 200$ GeV as in

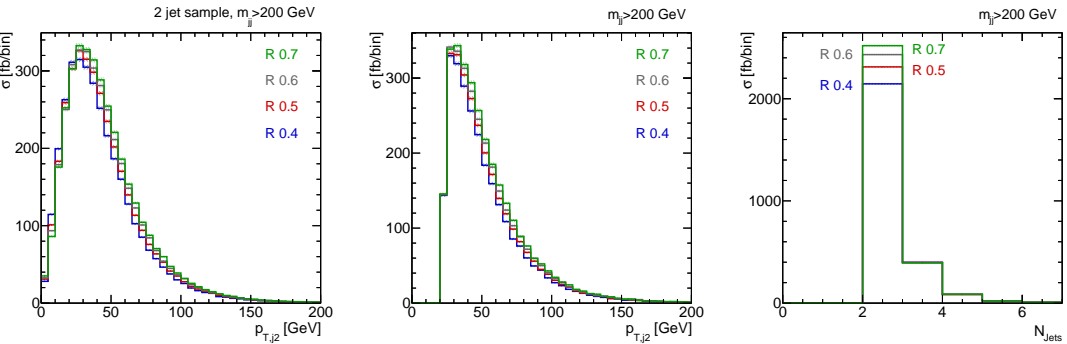

Figure 3: WBF signal distributions for the exclusive two-jet sample (left) and the full sample (center and right) requiring only $m_{jj} > 200$ GeV.

Eq.(1). For the different jet sizes we indeed find different $p_{T,j}$ spectra, for example for the second tagging jet. The larger we choose the tagging jet, the higher its transverse momentum becomes. The reason for this is simply that the tagging jet is more likely to pick up additional jet radiation, which by definition always increases its transverse momentum. Between $R = 0.4$ and $R = 0.7$ the peak of the $p_{T,j}$ spectrum shifts by around 5 GeV. Including a third merged hard jet reproduces this feature. Correspondingly, we also see that the number of events with two jets increases with the size of the tagging jets. Also the entire rate after the basic cut $m_{jj} > 200$ GeV increases, because the larger jet size helps the tagging jets to collect hadronic activity and pass this minimal $m_{jj}$ requirement. This (accidentally) leaves the number of three-jet and four-jet events constant.

Next, we show the actual $R$-dependence of the WBF signal and $Z$-background rates in the left panel of Fig. 4. To obtain results actually relevant for the analysis we require the pre-selection cuts of Eq. (3). For the QCD $Z$+jets backgrounds we also show a curve with up to four merged hard jets. The different lines correspond to two, three, and for the QCD background four hard jets, merged and combined with the parton shower. As expected, all rates increase with larger jet sizes. The question if we simulate additional hard jets or generate these jets with the parton shower plays essentially no role, independent of the nature of the hard process.

In the left panel of Fig. 4 the QCD $Z$+jets process shows a slightly larger slope than the electroweak signal and background processes. This can be explained by the higher rate of jet radiation off the QCD process and its external gluons. We can force the QCD and electroweak

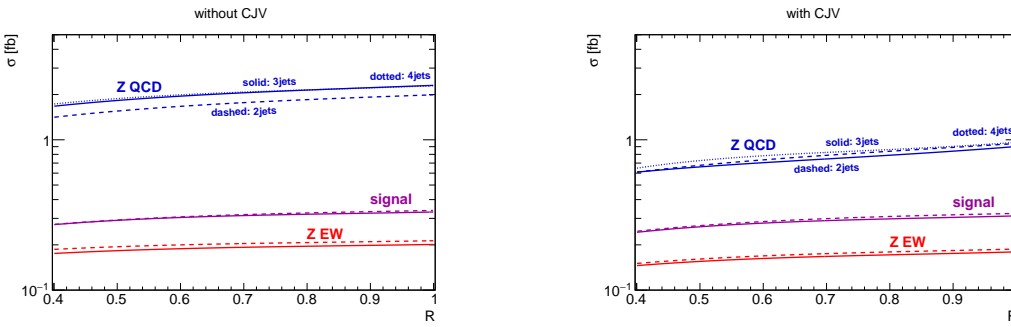

Figure 4: WBF signal and $Z$ background cross section dependence on $R$ without (left) and with (right) central jet veto. We always require the basic acceptance cuts of Eq.(3).

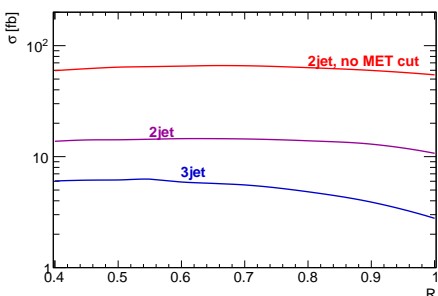

Figure 5: Dependence of the $(Z \to jj)H$ rate on $R$ with the basic cuts of Eq.(7).

processes into more similar setups by applying a central jet veto on jets with

$$p_{T,j_3} > 20 \text{ GeV} \qquad \min \eta_{j_{1,2}} < \eta_{j_3} < \max \eta_{j_{1,2}} \,. \qquad (5)$$

This way, the QCD $Z$+jets background will have a reduced jet activity in the central detector. What is left is jet radiation in the direction of the tagging jets. In the right panel of Fig. 4 we see the dramatic effect on the size of the QCD background. Beyond this, the $R$-dependence of the different signal and background processes is essentially identical.

Following this observation, the question arises whether the $R$-dependence of the signal and background cross sections is a feature of the WBF topology. As a test we analyze the process

$$pp \to ZH_{\text{inv}} \to jj\, H_{\text{inv}} \,, \qquad (6)$$

which has the same final state as our signal, but a different topology. We again merge up to three jets and use DELPHES3.3 for detector simulation. We apply the experimentally motivated cuts [21]

$$\begin{aligned} N_{\text{jets}} &= 2, 3 & p_{T,j_1} &> 45 \text{ GeV} & p_{T,j_{2,3}} &> 20 \text{ GeV} \\ \Delta R_{jj} &= 0.7 \dots 2.0 & m_{jj}(2\,\text{jets}) &= 70 \dots 100 \text{ GeV} \\ m_{jj}(3\,\text{jets}) &= 50 \dots 100 & \not{E}_T &= 120 \dots 160 \text{ GeV} \,. \end{aligned} \qquad (7)$$

The difference between the $ZH$ topology and the WBF topology is that, for the former, the two-jet system has a clear structure. First the invariant mass of the two jets is fixed, and second the geometric separation of the two jets can be related to the boost of the decaying $Z$-boson. In Fig. 5 we see that, for relatively small jets, the rates for two-jet events and three-jet events are stable. There are two features: first, the three-jet rate drops around $R = 0.55$. The corresponding topology is a boosted $Z$-boson recoiling against the missing momentum and a third jet, where the two $Z$-decay jets are boosted together into one observed jet. Second, the two-jet sample shows a mild drop towards even larger jet sizes. In this case the $Z$-boson recoils against the missing momentum alone and the point at which the two decay jets are merged is moved to larger jet sizes, $R$. We confirm this pattern by looking at the two-jet sample without the hard cut on $\not{E}_T$, making it even less likely that the two $Z$-decay jets are merged. Moreover, we observe that the relative growth of the cross section for the two-jet sample without the hard cut on $\not{E}_T$ is comparable to the one for the WBF signal.

Altogether, our comparison shows that the observed $R$-dependence of the WBF signal and background rates is not a specific feature of this process, but instead an effect which appears more generally, depending on the relevant phase-space regions. It is simply an effect of extra jet radiation, and we observed a much more distinctive $R$-dependence of the rate in the resonant $ZH$ topology.

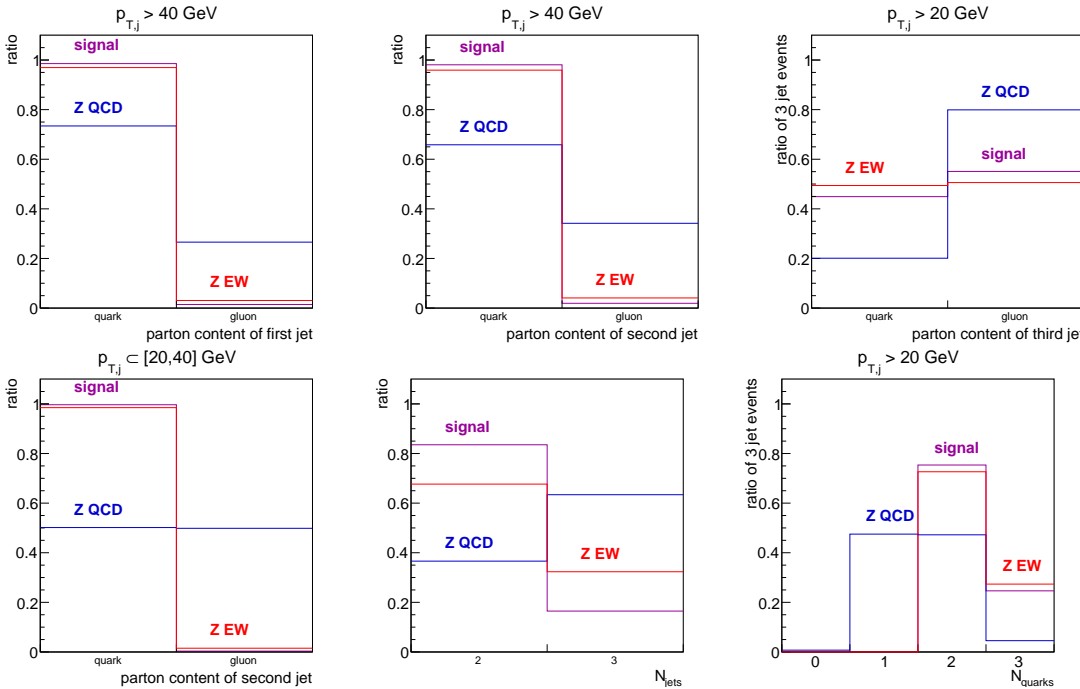

Figure 6: Parton content of the first, second, and third jet for WBF signal and the backgrounds, after the pre-selection cuts of Eq. (3). In addition, we show the parton content of the second jet in the slice $p_{T,j_2} = 20 \dots 40$ GeV, the number of jets, and the number of quarks in three-jet events.

## 4 Tagging jet content

After the pre-selection cuts of Eq. (3) it turns out that the largest backgrounds to invisible Higgs decays in WBF are clearly QCD processes radiating a weak boson, $V = W, Z$. Vetoing central jets is an established, but tedious, way to reduce these QCD backgrounds [8,9]. However, a simple veto is not necessarily the best use of the additional jet information. This motivates a more comprehensive analysis of the jet activity in the signal and the backgrounds [9]. In this paper we go a step further and test how we can use observables linked to quark vs gluon discrimination to control the QCD backgrounds [39,40].

Before we apply quark-gluon discrimination to the WBF signal and the QCD backgrounds we need to show that their partonic nature is indeed significantly different to motivate an application of quark-gluon discrimination. In general, we expect the tagging jets in the WBF signal to almost exclusively be quark-initiated, as confirmed in Fig. 6. For our parton-level illustration we do not apply a detector simulation, but we apply the pre-selection cuts of Eq.(3) with the exception of the $\not{E}_T$ cut. This cut is strongly affected by the detector simulation, and we replace it with a parton-level cut $p_T > 80$ GeV for the Higgs and the $Z$-boson. According

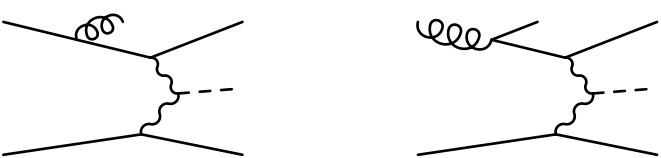

Figure 7: Three-jet contribution to the WBF signal with two or three quarks in the final state.

to the pre-selection cuts, a third jet has to be softer than the tagging jets. If the signal contains a third jet, it is equally likely to be a quark and gluon. Around 50 % of the events for which the third jet is a quark come from events with two quarks and one gluon where the gluon is harder than at least one of the quark jets, the other half is due to three quark events. The simple corresponding Feynman diagrams are given in Fig. 7.

For the electroweak $Z$+jets background all panels in Fig. 6 show essentially the same behavior as for the signal, with a slightly larger probability to see a third jet, reflecting the large number of topologies contributing to this background [11].

For the QCD $Z$+jets background the situation is completely different. Here, at tree level the harder tagging jet arise from a gluon in approximately 30% of the events. This fraction grows to around 35% for the second tagging jet and 80% for a possible third jet. From this estimate we expect the quark-gluon discrimination of the second jet to be most promising. However, the parton content also depends on the transverse momentum cut on the jet, as displayed in Fig. 6. Therefore, the discrimination power for the third jet should benefit from the lower $p_T$ threshold of this additional, central jet.

After establishing that quark-gluon discrimination can be useful to separate the WBF signal from QCD backgrounds we turn to appropriate observables. Some standard observables for quark vs gluon discrimination can be easily expressed in terms of particle flow (PF) objects or charged tracks as implemented in DELPHES3.3. They include [43–47]

$$n_{\mathrm{PF}} = \sum_{i_{\mathrm{PF}}} 1 \qquad\qquad C = \frac{\sum_{i_{\mathrm{PF}}, j_{\mathrm{PF}}} E_{T,i} E_{T,j} \left(\Delta R_{ij}\right)^{0.2}}{\left(\sum_{i_{\mathrm{PF}}} E_{T,i}\right)^2}$$

$$p_T D = \frac{\sqrt{\sum_{i_{\mathrm{PF}}} p_{T,i}^2}}{\sum_{i_{\mathrm{PF}}} p_{T,i}} \qquad\qquad Q^\kappa = \frac{\sum_{i_{\mathrm{trk}}} q_i \, p_{T,i}^\kappa}{\sum_{i_{\mathrm{trk}}} p_{T,i}^\kappa}$$

$$w_{\mathrm{PF}} = \frac{\sum_{i_{\mathrm{PF}}} p_{T,i} \, \Delta R_{i,\mathrm{jet}}}{\sum_{i_{\mathrm{PF}}} p_{T,i}} \ . \tag{8}$$

We define all observables except for $Q^\kappa$ on particle flow objects inside an anti-$k_T$ jet of size $R = 0.4$. In Fig. 8 we compare these for two idealized samples of exclusive QCD $Zjj$ events, namely events with either two quarks or two gluons in the final state. We apply a parton shower and the DELPHES3.3 detector simulation on the samples and only require the minimal selection cut $m_{jj} > 200$ GeV. We show the results for the second tagging jet in two slices

$$p_{T,j_2} = 20 \ldots 40 \,\mathrm{GeV} \qquad \text{and} \qquad p_{T,j_2} > 40 \,\mathrm{GeV} \ . \tag{9}$$

We observe clear differences between the quark samples and gluon samples in all distributions except for $Q^{1.0}$. Starting with $n_{\mathrm{PF},j_2}$, gluon jets clearly lead to more particle flow objects because they are more likely to radiate or split and therefore leave more tracks in the jet area. In addition, harder jets generally lead to more objects $n_{\mathrm{PF},j_2}$. The effect of the transverse momentum on $n_{\mathrm{PF},j_2}$ is significantly stronger than the difference between quarks and gluons. The variable $p_T D$ introduces a transverse momentum weight to maintain infrared safety, but after accounting for the inversion it follows a similar patterns as $n_{\mathrm{PF},j_2}$, with a reduced dependence on the transverse momentum of the jet.

Adding the angular distance to the jet axis in $w_{\mathrm{PF}}$ leaves the quark vs gluon separation similar to $n_{\mathrm{PF},j_2}$, but based on pure kinematics the softer jets now systematically reside at lower values of $w_{\mathrm{PF}}$. Again expanding $w_{\mathrm{PF}}$ to object-object correlators in $C$ almost de-correlates the quark-gluon discrimination from the transverse momentum of the jet. Finally, $Q^{1.0}$ is clearly not expected to be very useful, because all curves coincide for a large range of $Q^{1.0}$-values.

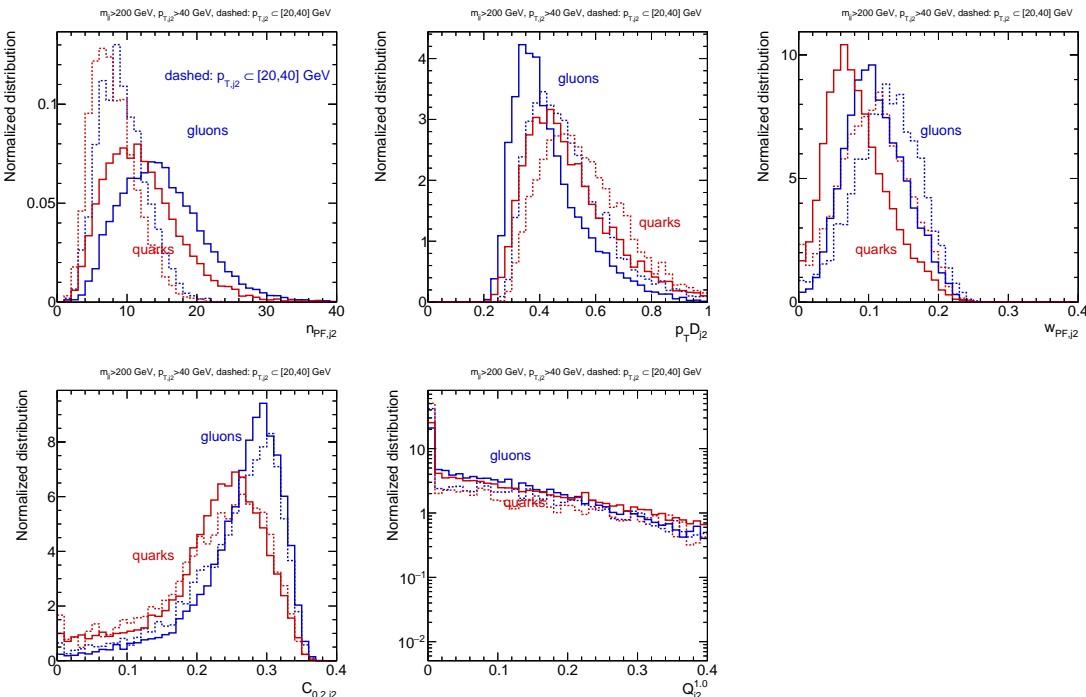

Figure 8: Distribution of the quark vs gluon discrimination variables listed in Eq.(8) for the second jet in a pure QCD $Zjj$ sample. We show partonic two-quark and two-gluon final states requiring $m_{jj} > 200$ GeV combined with the two slices $p_{T,j_2} = 20 \dots 40$ GeV and $p_{T,j_2} > 40$ GeV.

Given the similarity of the four leading distributions and the sizeable correlation with the transverse momentum suggests a dedicated multi-variate analysis to compare their performance.

Following our two findings, that WBF signal and QCD backgrounds differ in the partonic nature of the hard jet and that there are PF-level observables which can separate quarks and gluons, we now apply these observables to the fully simulated tagging jets. We employ the usual SHERPA simulation with two jets for the gluon fusion signal contribution and up to three hard jets merged for all other samples and a DELPHES3.3 fast detector simulation. Unlike for our preliminary results, the combination of shower and detector simulation no longer defines quark and gluon jets meaningfully. However, we expect some of the basic parton-level patterns to remain in the hadron-level final state. Again using anti-$k_T$ jets with $R = 0.4$ there is technically no problem in computing the PF observables defined in Eq.(8) after showering, hadronization and fast detector simulation.

We know that some of the PF distributions are strongly affected not only by the quark vs gluon nature of the jets, but also by its transverse momentum. In a realistic simulation of the WBF Higgs signal and the QCD $Z$+jets background both of these effects are present, Fig. 9. Note that, motivated by the central jet veto and in the absence of triggering requirements, the third, central jet can be as soft as $p_{T,j} = 20$ GeV. For the WBF signal where the tagging jet $p_T$-distribution peaks around $50 \dots 70$ GeV, the pre-selection cuts of Eq.(3) only have the mild effect of cutting off the second jet distribution below its peak. The difference for the QCD background is that typical jet radiation is neither forward nor at large transverse momentum. The combined tagging jet cuts of Eq.(3) extract events with harder tagging jets than we observe for the signal.

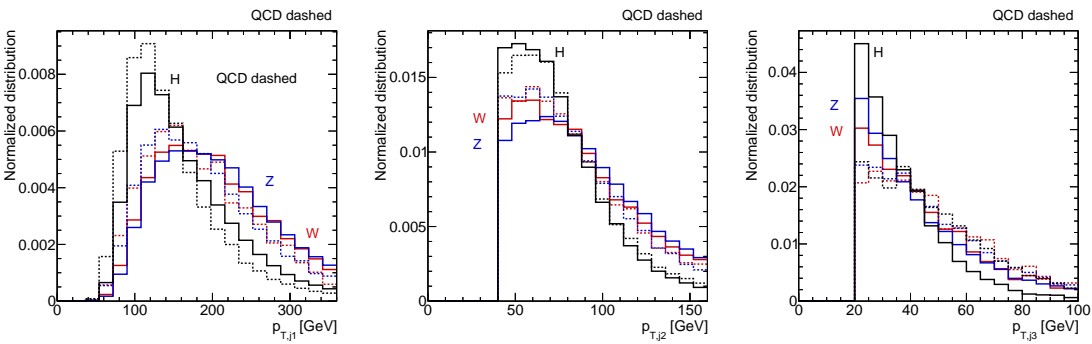

Figure 9: Transverse momentum distribution of the three jets for the WBF signal and the EW and QCD (dashed) $V$+jets background, after the pre-selection cuts of Eq.(3).

In our case the backgrounds are gluon-dominated and harder, so according to Fig. 8 the two effects strengthen each other for $n_{\mathrm{PF}}$, while they can lead to a compensation for $w_{\mathrm{PF}}$. Obviously, a proper multi-variate analysis can separate the different $p_T$ spectra from the parton nature. On the other hand, systematic uncertainties might well have a significant effect on this de-correlation for some of the PF observables.

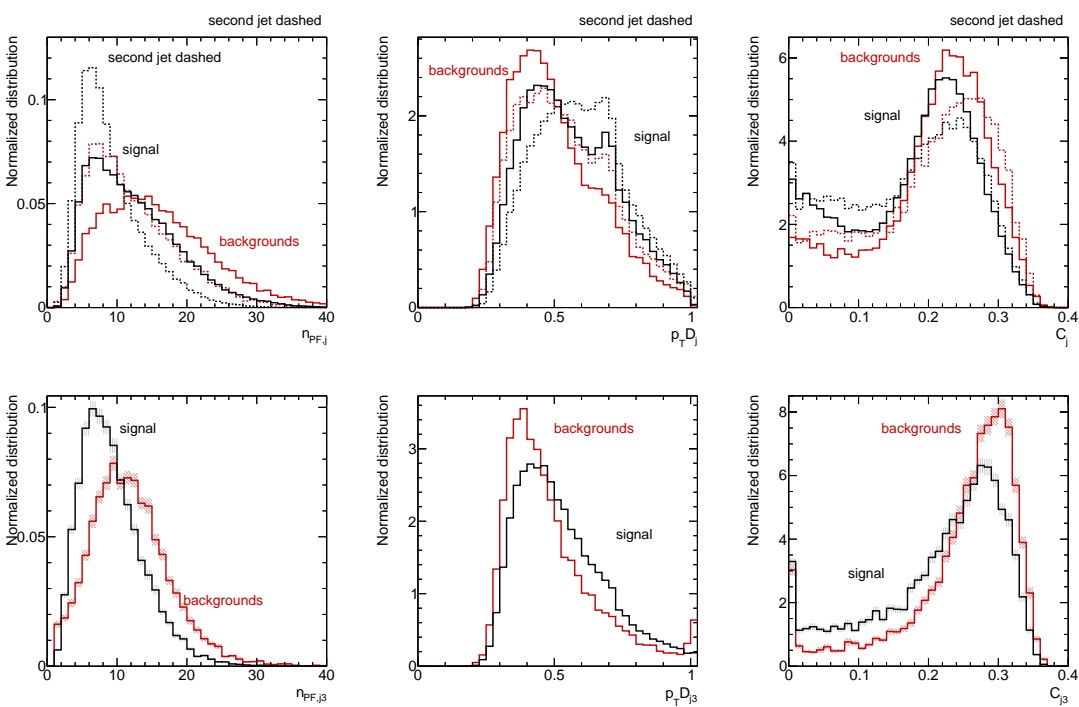

Figure 10: Distribution of the quark vs gluon discrimination variables $n_{\mathrm{PF}}$, $p_T D$, and $C$ for the WBF signal and the combined QCD and electroweak $V$+jets background, after the pre-selection cuts of Eq.(3). In the top panels we show the tagging jets, while in the bottom panels we show the softer third jet.

Finally, in Fig. 10 we display the leading three observables $n_{\mathrm{PF}}, p_T D$, and $C$ for the WBF signal including the contribution from gluon fusion and the combined $V$+jets background. The two tagging jets are displayed in the first row, while the second row shows the softer central jet, which often does not exist for the signal events. The pure number of PF objects

$n_{\mathrm{PF}}$ shows the best performance for the leading tagging jet, while $p_T D$ performs better on the second tagging jet. The reason for this is that the first tagging jet is significantly harder and hence includes more PF objects. The second peak in the $p_T D$ distribution for the signal and the electroweak backgrounds is an artifact of the pre-selection cuts.

For events with three jets we see that the $n_{\mathrm{PF}}$ distribution peaks at a value very similar as for the second tagging jet. This is accidental and an effect of the gluon nature of the jet pushing $n_{\mathrm{PF}}$ to larger values and the smaller momentum reducing $n_{\mathrm{PF}}$. Still $n_{\mathrm{PF}}$ as well as $C$ turn out to be promising observables for the WBF signal extraction.

## 5  Performance and triggering

Using the new handles developed in the last section, we still need to determine what their impact on the LHC reach for invisible Higgs decays is. First, we include the quark-gluon discrimination variables in a multi-variate analysis. We use boosted decision trees in TMVA [48, 49] with and without treating the quark-gluon discrimination variables as input parameters for the classification. We use the AdaBoost algorithm with 70 trees, a maximum depth of 3 and require a minimum node size of 5 % of the number of events. The input observables for the BDT analyses are listed in Tab. 1.

Table 1: Sets of variables used for the BDT analysis. Variables with the subscript $jj$ refer to the two tagging jets.

| Set | Variables |
|---|---|
| jet-level $j_1$, $j_2$ | $p_{T,j_1}$    $p_{T,j_2}$    $\Delta\eta_{jj}$    $\Delta\phi_{jj}$    $m_{jj}$    $\slashed{E}_T$    $\Delta\phi_{j_1,\slashed{E}_T}$    $\Delta\phi_{j_2,\slashed{E}_T}$ |
| subjet-level $j_1$, $j_2$ | $n_{\mathrm{PF},j_1}$    $n_{\mathrm{PF},j_2}$    $C_{j_1}$    $C_{j_2}$    $p_T D_{j_1}$    $p_T D_{j_2}$ |
| $j_3$ angular information | $\Delta\eta_{j_1,j_3}$    $\Delta\eta_{j_2,j_3}$    $\Delta\phi_{j_1,j_3}$    $\Delta\phi_{j_2,j_3}$ |
| jet-level $j_1$-$j_3$ | jet-level $j_1$, $j_2$   +   $j_3$ angular information   +   $p_{T,j_3}$ |
| subjet-level $j_1$-$j_3$ | subjet-level $j_1$, $j_2$   +   $n_{\mathrm{PF},j_3}$    $C_{j_3}$    $p_T D_{j_3}$ |

In Fig. 11, we display the signal efficiency vs inverse background efficiency (ROC curve) for different sets of BDT input variables. The left panel of Fig. 11 shows the results from a BDT using the full set of standard WBF variables, containing the jet-level ($p_T$, $\eta$, $\phi$) information on the tagging jets and a possible third jet, provided $p_{T,j_3} > 20$ GeV. We compare it to a BDT which in addition uses the subjet-level information $n_{\mathrm{PF}}$, $C$, $p_T D$. In both cases, the variable most often used for the splittings of the BDT is $\Delta\eta_{j_1,j_3}$. The most important subjet-level variables are the number of constituents and the $p_T D$ of the third jet, individually ranked after the angular separation variables of the jets. This ranking corresponds to the observation that according to the left panel the subjet-level observables do not lead to a visible improvement of the classification power. A similar picture emerges from the center panel, where we show the classification power based on the two tagging jets alone. Again, we separate jet-level information alone from the combined jet-level and subjet-level information. For the tagging jets the most important single variable comes out to be $\Delta\phi_{jj}$, while the most important subjet-level variable is $n_{\mathrm{PF},j_1}$, ranked fifth. The limited impact of the new set of subjet-level observables in WBF Higgs production is explained by the large number of observables shown in Tab. 1. Even for a $2 \to 3$ process, the information on the event kinematics is eventually saturated.

From a recent study, we know that the only way to still increase the performance of the WBF analysis is to use information on softer central jets [9]. The issue with soft central jets

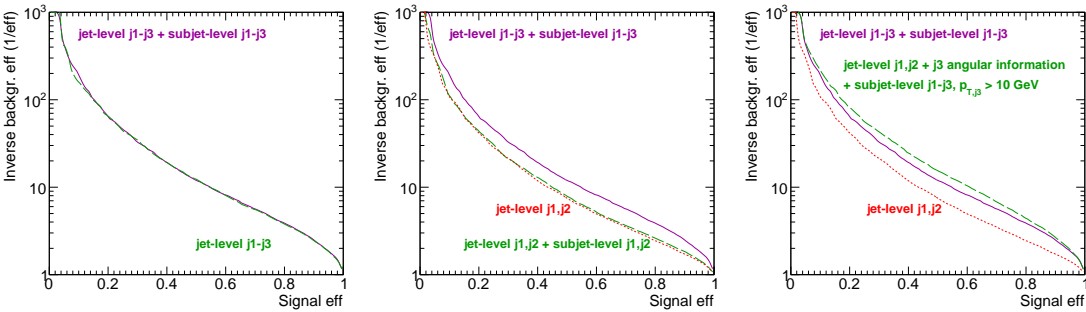

Figure 11: Signal efficiency vs inverse background efficiency based on jet-level and additional subjet-level information of the leading three jets (left). In the center panel we show the jet-level and subjet-level of the tagging jets only, while to the right we include the subjet-level information on the third jets with a lower threshold of $p_{T,j_3} > 10$ GeV.

is that they are hard to calibrate. One way around this limitation is to consider a soft third jet merely a container for subjet-level observables. In the right panel of Fig. 11 we show the performance once we include jet-level and subjet-level information on a soft third jet, but omit the jet-level $p_{T,j_3}$. In this scenario, the subjet-level discrimination variables can significantly improve the analysis, with the most important variable becoming $n_{\mathrm{PF},j_3}$. We have checked that adding the information on the jet-level $p_{T,j_3}$ to the input variables does not further increase the performance of the BDT. Including all particle flow observables from Eq.(8) in this analysis is likely overly optimistic. A proper tracker-focussed analysis with dedicated tunes of simulation tools is beyond the reach of this first theoretical study, but our results indicate that such an in-depth analysis is promising.

For the HL-LHC, a major issue for WBF Higgs analyses in general and invisible Higgs searches in particular will be detector limitations and trigger thresholds. Using the pre-selection cuts of Eq.(3) is likely overly optimistic. To systematically compute the expected limit on invisible Higgs decays for a multi-variate analysis we use an implementation of the CLs method [50] in CHECKMATE [51]. First, we scan through different cuts of the BDT classifier variable and obtain the corresponding signal and background efficiencies. Taking into account the cross section after our pre-selection cuts of Eq.(3) we can calculate the 95 % CLs limit for each point on the ROC curve and obtain the best limit. We assume a systematic uncertainty of 3%.

In Fig. 12, we show the effect of a variation of the triggers on missing energy, the invariant mass and the transverse momentum of the tagging jets. As a reference value we also show the expected results from the leptonic $ZH$ analysis from the Appendix. While a variation of the minimum missing energy or transverse momentum has a big impact on the reach for invisible Higgs decays, the sensitivity seems is largely independent of minimum invariant mass of the tagging jets, closely related to their rapidity separation.

We find that a trigger cut of $\not{E}_T > 200 \dots 300$ GeV reduces the sensitivity to invisible Higgs decays by roughly a factor of $1.5 \dots 2.7$. While we show trigger-level cuts below $\not{E}_T > 140$ GeV, we need to keep in mind that in this regime we will likely have to take into account more background processes.

The trigger cut on the transverse momentum of the tagging jets has equally strong impact on the WBF search, as can be seen already in Fig. 9. Increasing the requirement to around $p_{T,j_{1,2}} > 130$ GeV leads to a drop of the WBF sensitivity to below the $ZH$ reference point.

Finally, the variation of the $m_{jj}$ trigger has almost no impact on the sensitivity of the analysis. Even for a stiff cut $m_{jj} > 2500$ GeV, the reach in terms of the invisible Higgs branching

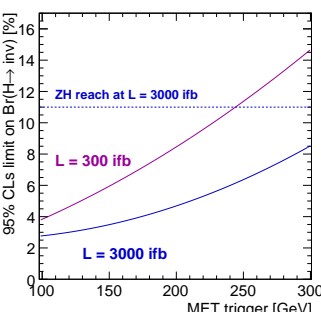 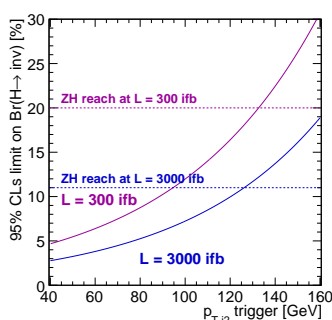 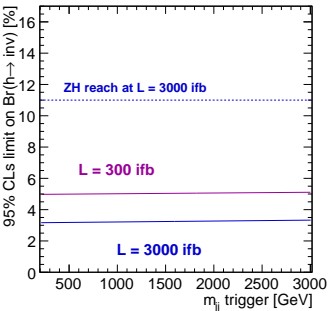

Figure 12: CLs limits on invisible Higgs decays from weak boson fusion, as a function of trigger cuts on missing transverse energy (left), the transverse momentum of the tagging jets (center), and the invariant mass of the tagging jets (right). As a reference we also display the reach in the leptonic $ZH$ channel, described in the Appendix.

ratio stays at the 5% level for an integrated luminosity of $300\,\text{fb}^{-1}$ and and at 3% for $3000\,\text{fb}^{-1}$. Since $\Delta\eta_{jj}$ is strongly correlated with the invariant mass of the tagging jets, we confirm that an increase in terms of $\Delta\eta_{jj}$ leaves the sensitivity almost unchanged up to $\Delta\eta_{jj} > 7$.

## 6 Outlook

Higgs production in weak boson fusion is the leading process to search for invisible Higgs decays. The unique tagging jet signature offers many handles to suppress the different backgrounds, either based on the actual tagging jets or based on the global QCD structure of the events.

In this paper we have studied several aspects of WBF tagging jet analyses. First, we find that backgrounds can indeed be controlled, but that eventually single top production might limit the correspondence of $W$+jets and $Z$+jets backgrounds. Next, we have studied how the dependence of the perturbative signal cross section prediction on the size of the tagging jets is not specific to WBF processes and subject to the details of the analysis.

In the main part of the paper we systematically introduced subjet-level observables for the two tagging jets as well as for possible central jets. We find that this additional subjet-level information can be used to suppress backgrounds and increase the purity of an event sample, but that it does not significantly improve a multi-variate analysis based on the complete available jet-level information. The reason is that the kinematics of the signal and background processes is already fully constrained by the jet-level analysis. However, we find that subjet observables will allow us to rely less on notorious observables, like a central jet veto survival probability. Moreover, subjet information and in particular tracking information should eventually allow us to exceed the jet-level performance. Higgs production in weak boson fusion gives us a strong case to consistently consider jets not as the main analysis objects, but as containers including valuable subjet-level information.

Finally, we have studied how increased trigger and detector thresholds will harm the search for invisible Higgs decays at the HL-LHC. Transverse momentum thresholds can seriously reduce the reach for invisible branching ratios from the 2% level to the 10% level, i. e. the typical reach of the leptonic $ZH$ channel. This is especially true for the subleading tagging jet. Relying, for example, on the invariant mass of the tagging jets, in contrast, has hardly any effect on our WBF analysis.

## Acknowledgements

The authors would like to thank Monica Dunford and especially Manuel Geisler for very useful discussions about the discrimination power of the third jet and for their assistance with TMVA. We thank Yacine Haddad for pointing out that the variable $p_T D$ is widely used in the experimental community. AB acknowledges support by the IMPRS-PTFS. AB and TP are supported by the DFG Forschergruppe *New Physics at the LHC* (FOR 2239).

## A   Appendix: ZH benchmark

To make a meaningful statement about the impact of new approaches on invisible Higgs searches in weak boson fusion, we need a benchmark. We therefore compute the prospective reach of the associated production channel

$$pp \to ZH_{\text{inv}} \to \ell^+\ell^- H_{\text{inv}} \tag{10}$$

at the HL-LHC. The signature with two same-flavor opposite-sign (SFOS) leptons plus missing energy is experimentally much more stable than the WBF tagging jets. Note that the loop-induced gluon fusion channel can have sizeable impact on the sensitivity of this channel [14]. We generate both, tree-level quark-induced and loop-level gluon-induced events at 14 TeV using SHERPA and DELPHES3.3. We also use OPENLOOPS for the loop calculations in the gluon fusion channels.

The main backgrounds are quark-induced and gluon-induced $Z_{\ell\ell}Z_{\nu\nu}$ production. Other important backgrounds are $WZ$ production with a missing lepton from the $W$ decay, $WW$ production where the invariant mass of the leptons comes out close to the $Z$ mass, and leptonic $t\bar{t}$ production. We generate these background processes using SHERPA and include a loop-level sample for the irreducible gluon-induced $ZZ$ background using OPENLOOPS. All total rates are normalized to their respective NNLO predictions [52]. We require the cuts

$$p_{T,\ell_1} > 26\,\text{GeV} \qquad p_{T,\ell_2} > 7\,\text{GeV} \qquad \eta_e < 2.47 \qquad \eta_\mu < 2.5$$
$$|m_{\ell\ell} - m_Z| < 5\,\text{GeV} \qquad \Delta R_{\ell\ell} < 1.8 \qquad \slashed{E}_T > 60\,\text{GeV} \qquad \Delta\phi(p_T^{\ell\ell}, \slashed{E}_T) > 2.7 \,. \tag{11}$$

Because of the simple $2 \to 2$ kinematics we do not expect a large benefit from using a BDT compared to a cut-and-count analysis. However, to compare the results to our WBF analysis we also analyze the $ZH$ channels using the BDT implementation of TMVA. In addition to the variables given in Eq. (11) we include the observables

$$\left\{ \eta_{\ell_1}, \, \eta_{\ell_2}, \, \phi_{\ell_1}, \, \phi_{\ell_2}, \, \phi_{\slashed{E}_T}, \, \frac{p_T^{\ell\ell}}{m_T}, \, N_{\text{leptons}}, \, N_{\text{jets}} \right\} \,. \tag{12}$$

Table 2: 95 % CLs limits on the invisible Higgs branching ratio from the leptonic $ZH$ channel. The ATLAS indicated by an asterisk result is taken from Ref. [21].

| Systematics | Luminosity [fb$^{-1}$] | | | |
|---|---|---|---|---|
| | 36.1 [21] | 36.1 | 300 | 3000 |
| 1% sys. | 39% | 39% | 17% | 8% |
| 2% sys. | | 43% | 20% | 11% |

The resulting 95 % CLs limits for a systematic uncertainty of 1% and 2% are summarized in Table 2. While we assume at best a 2% uncertainty to be realistic, we also show the results for a 1% uncertainty as a reference. The comparison with the expected ATLAS limit [21] at 13 TeV indicates that appropriate data-driven background rejection techniques can compensate for otherwise large systematics. The main factor is the normalization of the leading $ZZ$ background, where we apply a global $K$-factor to account for the NNLO correction [52], while ATLAS uses bin-wise factors for $m_{ZZ}$ [53].

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
