# Peer review of "Tagging Jets in Invisible Higgs Searches"

_SciPost Physics, doi:SciPost Phys. 4, 035 (2018)_

## Round 3 · Referee Report · Anonymous · 2018-2-14

Strengths

see the main report below

Weaknesses

see the main report below

Report

The article discusses two questions for invisible Higgs searches in
weak-boson fusion topologies.

The first question concerns the impact of changing the radius of the
jet. The results are not overly surprising. Nevertheless, they can
provide a useful reference for future studies, e.g. of using different
jet radii to help probe and constrain systematic uncertainties.

The main interesting part of the paper concerns the value of applying
quark-gluon discrimination jet-substructure variables to invisible
Higgs searches. The authors' surprising finding is that this brings
almost no advantage (Fig.11), despite the fact that the quark/gluon
composition of signals v. backgrounds is somewhat different (Fig.6).

This is a finding that deserves to be shared and ultimately I
recommend that the paper be published. However first I recommend that
the authors address the points below.

Requested changes

1- The abstract mentions the importance of controlling single-top
backgrounds. However if one examines a CMS analysis of this channel
1610.09218 (Ref.[19]), Fig.3, they have barely any background from top
production. The authors should adapt and/or clarify their statements
in this regard.

2- In the description on p.5 of their simulation setup, some parts
are carried out at LO (WBF signal), while others appear to be at NLO
(given the reference to OpenLoops), such as gluon-fusion Higgs + 2
jets. It would be good to be explicit about the use of NLO for the
latter. While for the former, I think it would be useful if the
authors gave an indication of the expected effects of (N)NLO
corrections, since these are known in many cases.

3- p.7, two lines before Eq.(5), there's a typo "the the".

4- On p.7, the authors state that between $R=0.4$ and $R=0.7$ the
peak of the $p_{T,j}$ spectrum shifts by around 10 GeV. But in Fig.3
it looks closer to 5 GeV.

5- in Eq. (8) (definition of substructure variables), the
definition of $p_TD$ looks wrong. In the numerator of the first
equation to the right of the equal sign the $p_{T,i}$ terms should
appear squared, which also prevents the simplification in the
right-most part of the equation.

6- Still for Eq.(8) it would be better if Refs.[41,42] were
replaced with references to the original articles that introduced
the individual variables (rather than conference proceedings that
just use them).

7- There's a typo in the last main sentence of the first big
paragraph on p. 14: "even" -> "event"

8- Relating to Fig.11(right), and also the statement in the
abstract that subjet information [...] can be used to relieve some
of the pressure on other critical observables: by this, the authors
presumably refer to the fact that they've obtained the improvement
in performance relative to the left/middle figures, despite the fact
that while lowering the 3rd jet cut to 10 GeV, they've not used the
3rd jet $p_T$ as an input. However, I think an important comparison in this
regard would be with a situation where they keep the lower $p_T$ cut
and use the 3rd jet $p_T$ as an input (with and without jet
substructure variables). I encourage the authors to add this.

  • validity: high
  • significance: good
  • originality: good
  • clarity: good
  • formatting: excellent
  • grammar: perfect

Author:  Anke Biekoetter  on 2018-04-25

(in reply to Report 1 on 2018-02-14)

Dear referee,

Thank you for the useful corrections.

We have looked through your suggestions and modified the paper accordingly:

1- The abstract mentions the importance of controlling single-top backgrounds. However if one examines a CMS analysis of this channel 1610.09218 (Ref.[19]), Fig.3, they have barely any background from top production. The authors should adapt and/or clarify their statements in this regard. -> The increased contribution from single top events that we see is due to the more inclusive signal definition used for machine learning approaches. However, we understand that the sentence could be misleading and therefore we have changed it from 'First, precision analyses of the central jet activity require full control of single top production.' to 'First, precision analyses of the central jet activity require full control of single top production in some analyses.'

2- In the description on p.5 of their simulation setup, some parts are carried out at LO (WBF signal), while others appear to be at NLO (given the reference to OpenLoops), such as gluon-fusion Higgs + 2 jets. It would be good to be explicit about the use of NLO for the latter. While for the former, I think it would be useful if the authors gave an indication of the expected effects of (N)NLO corrections, since these are known in many cases. -> The description of the simulation setup was unclear before. We modified the simulation setup description to clarify that our simulation is LO and OpenLoops was only used for loop-induced LO processes. We did not consider (N)NLO effects and feel that they are better left to a dedicated study.

3- p.7, two lines before Eq.(5), there's a typo "the the". -> fixed

4- On p.7, the authors state that between R=0.4 and R=0.7 the peak of the pT,j spectrum shifts by around 10 GeV. But in Fig.3 it looks closer to 5 GeV. -> fixed

5- in Eq. (8) (definition of substructure variables), the definition of pTD looks wrong. In the numerator of the first equation to the right of the equal sign the pT,i terms should appear squared, which also prevents the simplification in the right-most part of the equation. -> Thank you for pointing out this massive typo. However, the definition of pTD was only wrong in the text, not in the implementation of the variable. We have fixed the defintion in the text.

6- Still for Eq.(8) it would be better if Refs.[41,42] were replaced with references to the original articles that introduced the individual variables (rather than conference proceedings that just use them). -> We agree. The references were replaced by references to the original articles.

7- There's a typo in the last main sentence of the first big paragraph on p. 14: "even" -> "event" -> fixed

8- Relating to Fig.11(right), and also the statement in the abstract that subjet information [...] can be used to relieve some of the pressure on other critical observables: by this, the authors presumably refer to the fact that they've obtained the improvement in performance relative to the left/middle figures, despite the fact that while lowering the 3rd jet cut to 10 GeV, they've not used the 3rd jet pT as an input. However, I think an important comparison in this regard would be with a situation where they keep the lower pT cut and use the 3rd jet pT as an input (with and without jet substructure variables). I encourage the authors to add this. -> While preparing the paper this comparison was completed. However, there was no (visible) difference between including the 3rd jet pT as an input or not. Therefore we did not feel like it would add to the publication.

Thank you again for your helpful corrections.

Yours faithfully, Anke Biekoetter (on behalf of the authors)

---

## Round 4 · List of Changes

• corrected typos
  • clarified perturbative order of event simulation (LO)
  • added a sentence on the possible influence of the pT of the third jet on the significance of Higgs to invisibles

---

## Editorial Decision

published